# The MinCDE Cell Division System Participates in the Regulation of Type III Secretion System (T3SS) Genes, Bacterial Virulence, and Motility in *Xanthomonas oryzae* pv. *oryzae*

**DOI:** 10.3390/microorganisms10081549

**Published:** 2022-07-31

**Authors:** Yichao Yan, Yanyan Wang, Xiaofei Yang, Yuan Fang, Guanyun Cheng, Lifang Zou, Gongyou Chen

**Affiliations:** 1Shanghai Collaborative Innovation Center of Agri-Seeds, School of Agriculture and Biology, Shanghai Jiao Tong University, Shanghai 200240, China; yichao.yan@sjtu.edu.cn (Y.Y.); tsl-wangyy@taslybio.com (Y.W.); yangxiaofei6100@163.com (X.Y.); floragreen@sjtu.edu.cn (Y.F.); cgy52494@outlook.com (G.C.); gyouchen@sjtu.edu.cn (G.C.); 2State Key Laboratory of Microbial Metabolism, School of Life Sciences & Biotechnology, Shanghai Jiao Tong University, Shanghai 200240, China

**Keywords:** *Xanthomonas oryzae* pv. *oryzae*, MinCDE system, type III secretion system, virulence, motility

## Abstract

*Xanthomonas oryzae* pv. *oryzae* (*Xoo*) causes bacterial leaf blight (BLB) in rice, which is one of the most severe bacterial diseases in rice in some Asian countries. The type III secretion system (T3SS) of *Xoo* encoded by the hypersensitive response and pathogenicity (*hrp*) genes is essential for its pathogenicity in host rice. Here, we identified the Min system (MinC, MinD, and MinE), a negative regulatory system for bacterial cell division encoded by *minC*, *minD*, and *minE* genes, which is involved in negative regulation of *hrp* genes (*hrpB1* and *hrpF*) in *Xoo*. We found that the deletion of *minC*, *minD*, and *minCDE* resulted in enhanced *hrpB1* and *hrpF* expression, which is dependent on two key *hrp* regulators HrpG and HrpX. The *minC*, *minD*, and *minCDE* mutants exhibited elongated cell lengths, and the classic Min system-defective cell morphology including minicells and short filamentations. Mutation of *minC* in *Xoo* resulted in significantly impaired virulence in host rice, swimming motility, and enhanced biofilm formation. Our transcriptome profiling also indicated some virulence genes were differentially expressed in the *minC* mutants. To our knowledge, this is the first report about the Min system participating in the regulation of T3SS expression. It sheds light on the understanding of *Xoo* virulence mechanisms.

## 1. Introduction

*Xanthomonas* is a genus of Gram-negative bacteria that includes numerous species that cause disease in over 400 different plant hosts, including rice, citrus, wheat, cabbage, tomato, cassava, and pepper [1]. *Xanthomonas oryzae* pv. *oryzae* (*Xoo*) is widespread in Southern China, West Africa, and Southeast Asian countries such as Thailand and Vietnam [2]. The phytopathogenic *Xoo* infects rice, causing bacterial leaf blight (BLB), which induces worldwide output losses of up to 50% [3]. *Xoo* produces a variety of virulence factors, including lipopolysaccharides (LPS), exopolysaccharides (EPS), extracellular enzymes, toxins, adhesions, and effectors injected into host rice by the type III secretion system (T3SS), and so on [4]. The *Xoo* T3SS that controls the pathogenicity in susceptible host rice is encoded by a hypersensitive response and pathogenicity (*hrp*) gene cluster, which contains 27 genes including 10 *hrp*, 9 *hrc* (*hrp* conserved), and 8 *hpa* (*hrp*-associated) genes [5].

The expression of *hrp* genes of *Xoo* is significantly stimulated in planta, or in minimum medium (XOM3), an artificial *hrp*-inducing medium, but inhibited in the nutrient-rich medium [6]. The expression of *Xoo hrp* genes was regulated by two key regulators, HrpG and HrpX. HrpG belongs to the OmpR-family response regulator of two-component regulatory systems. It has a response receiver (RR) domain at the N-terminus and a DNA-binding motif at the C-terminus [7]. HrpX binds directly to the plant-inducible promoter (PIP) box consensus motif (TTCGC-N15-TTCGC), a cis-regulatory region [8]. HrpG acts as a positive regulator of *hrpX* expression and is also a crucial regulator in some *Xanthomonas* species or pathovars. In *X*. *campestris* pv. *campestris* (*Xcc*), HpaS has been demonstrated to act as a sensor kinase for HrpG; however, there is no intact homolog of the *hpaS* gene of *Xcc* in the genomes of *X**oo* strains [9]. In *Xcc 8004*, the sensor kinase RpfC can regulate *hrpX* and T3SS genes expression in the nutrient broth and the host environment via the DSF cell–cell communication system [10]. The global transcriptional regulator Clp has been reported to bind the promoter regions of downstream targets *zur*, cellulases *engXCA*, and *fhrR*, hence promoting the production of virulence-associated genes [11]. In *X*. *citri* subsp. *citri* (*Xcci*) [12], Lon, an ATP-dependent protease, can degrade HrpG protein in the rich medium; however, Lon was phosphorylated and lost its inhibitory impact on HrpG in host plants [13]. Lon inhibits the expression of T3SS and flagellar synthesis and participates in cell division and exopolysaccharide formation [13]. In *Xoo*, several components, including GntR-family regulator Trh [14], and the two-component systems PhoP/PhoQ [15], have been associated with *hrpG* expression. Moreover, KdgR, a negative regulator of *hrpG*, has been reported to directly bind to the promoter regions of *hrpG*, thereby repressing the transcription of *hrp* genes [16]. The other upstream regulators of T3SS in *Xoo* remain unknown and need further investigation.

How a cell finds its middle has been studied for the last 50 years in *Escherichia coli* [17]. Two negative regulatory systems for cell division have been identified in *E*. *coli*. One is the nucleoid occlusion (NO) system that prevents Z-ring formation over the nucleoid, and the other is the Min system encoded by *minC*, *minD*, and *minE* genes, which inhibits the formation of Z-ring at the poles [18]. A current model suggests that the concentration gradient of MinC in a cell regulates the Z-ring position [19]. MinC is an inhibitor of FtsZ and can directly interact with FtsZ, thereby inhibiting its polymerization [20,21,22]. MinD is an ATPase that can bind and recruit MinC to the membrane [23]. MinE can stimulate the ATPase activity of MinD, and thus detach it from the membrane [24]. As MinE assembles at mid-cell, and cycles back and forth toward the cell poles, the dissociation of the MinC/MinD complex results in the oscillation behavior of Min proteins in cells [19,22,25]. This causes a concentration gradient of MinC/MinD complex to be highest at the cell poles and lowest at mid-cell, thus allowing Z-ring formation at mid-cell in a narrow zone [19]. The Min system-defective mutants share similar phenotypic characteristics: minicells and filamentous cells [17]. The focus of past studies on the role of the Min system was to characterize its oscillation and interaction with divisome-associated proteins.

Some current studies have suggested the involvement of the Min system in cellular processes such as bacterial motility, colonization, and virulence. The *minC* mutants of *Proteus mirabilis* and *Helicobacter pylori* significantly reduced swarming motility [26,27]. *Neisseria gonorrhoeae* (*Ng*) mutants without MinD or MinC exhibited decreased adherence to urothelial cells [28]. MinCD complex of *E*. *coli* can attach to the membrane and assist in segregating chromosomes [29]. The MinC oscillations from pole to pole also were observed in *Xcci*, and similar to Min system-defective mutants in *E*. *coli*, the *Xcci minC* mutant could form branching cells with aberrant extension and bulging at both poles [30]. A current finding identified MinD of *Xoo* as a host-induced protein required for *Xoo* full virulence in host rice [31], indicating that some mechanisms and pathways of *Xanthomonas* Min proteins that regulate virulence during infection are unknown.

In this study, we screened two transposon mutants, 8–24 and 24–46, with up-regulated expression of *hrpF* and *hrpB1,* using a Tn5 transposon mutagenesis, in which the transposon was inserted into the *Xoo* PXO99^A^
*minC* and *minD* genes, respectively. The *Xoo* PXO99^A^ Min system is composed of MinC, MinD, and MinE proteins that are encoded by the *minC* (*PXO_04463*), *minD* (*PXO_04464*), and *minE* (*PXO_04465*) genes, respectively. We revealed the link between the Min system and the T3SS expression of *Xoo*. We demonstrated the negative effects of the Min system on T3SS expression through the HrpG–HrpX regulatory pathway, and the involvement of the Min system in *Xoo* cell division, full virulence, swimming motility, and biofilm formation. Our findings propose new indications that the Min system contributes to the virulence regulatory networks of *Xoo*.

## 2. Materials and Methods

### 2.1. Bacterial Strains, Plasmids, and Growth Conditions

*Xoo* wild-type PXO99^A^ [32] and other Xoo strains were grown on nutrient-rich NA plates, or in NB medium and minimal XOM3 medium with appropriate antibiotics at 28 °C [6,33]. *E. coli* strains were cultured on LB agar (LA) plates or in Luria-Bertani (LB) medium with appropriate antibiotics at 37 °C. The following final concentrations of antibiotics were used: kanamycin (Km), 50 μg/mL; gentamicin (Gm), 25 μg/mL; spectinomycin (Sp), 100 µg/mL. The bacterial strains and plasmids employed in this study are listed in Appendix A.

### 2.2. Construction of Mutant and Complementation Strains

All primers used in this study are listed in Appendix A. We constructed the deletion mutant of the Min system (*minC*, *minD*, and *minCDE*) using the suicide vector pKMS1 with *sacB* gene by homologous recombination [34]. The specific primers amplified the upstream, and downstream sequences of *minC*, *minD*, and *minCDE* were ligated into the pKMS1 to create the pK–min-system construct. The plasmids were transformed into the *Xoo* wild-type PXO99^A^ by electroporation, respectively. The colonies were followed by the selection on NA plates with 10% sucrose. The PΔ*min*-system deletion mutants were selected by the sensitivity to Km on the NA medium. The transposon mutant strains 8–24 and 24–46 with highly *hrpF* and *hrpB1* expression were selected, respectively. For complementation of the PΔ*min*-system, the fragments containing the *minC*, *minD*, and *minCDE* encoding regions were amplified using the primer pairs (Appendix A) and cloned into pML123 to obtain the recombinant plasmids pML123-*minC*, pML123-*minD*, and pML123-*minCDE*, respectively. Electroporation of the recombinant plasmids was transformed into the insertion or deletion mutants to obtain the complemented strains CPΔ*minC*, CPΔ*minD*, and CPΔ*minCDE*.

### 2.3. Synteny Analysis on Chromosomes

To determine whether Min system genes are conserved among *Xanthomonas* strains, we use the SyntTax bioinformatics tool with the ABSYNTE algorithm to perform the synthetic analysis (http://archaea.u-psud.fr/synttax/, accessed on 5 March 2022) [35], employing 10 *Xanthomonas* genomes as a reference, including *Xoo* PXO99^A^ and PXO86, *X*. *oryzae* pv. *oryzicola* BLS256 and RS105, *X*. *axonopodis* pv. *commiphoreae* LMG26789, *X*. *vasicola* NCPPB902, *X*. *citri* subsp. *citri* 49, 29–1, and 306, and *X*. *campestris* pv. *campestris* 8004.

### 2.4. Microscopy

Wild-type PXO99^A^ and the Min mutant strains were grown on NA plates at 28 °C. Cells were washed twice with PBS and fixed to 1.5 mL tubes with 3% glutaraldehyde overnight at 4 °C. After removing the blocking buffer and washing twice with PBS, the cells were stationary for 1 h with 1% osmic acid at 4 °C. Then, the bacterial cells were dehydrated with ethyl alcohol concentration and placed into a drying oven overnight at 37 °C for CO_2_ drying. The microstructures of cells were observed by using scanning electron microscopy (SEM). We utilized the ImageJ software to measure the lengths of the cells. For transmission electron microscope (TEM) analysis, the experiment was employed according to our previous protocol [36]. For fluorescence microscope observation, the *Xoo* strains carrying pHM1-*gfp* with a highly expressed GFP were analyzed according to our previous protocol [36].

### 2.5. Biofilm Formation Assay

Biofilm formation assay was determined as described previously [36]. *Xoo* strains were grown for 12 h in NB medium and diluted with 1:100 to overnight culture in NB medium. The bacterial cells were collected by centrifugation at 5000 rpm for 3 min. Then, we adjusted the optical density (OD) from 600 nm to 2.0 and incubated 5 mL bacterial suspension in a test tube at 28 °C. Following three days of incubation, the supernatant was carefully removed, and the adhering bacterial cells were stained with 6 mL 0.1% crystal violet (CV) for 30 min. The CV-stained cells were washed twice with distilled water and dried in a 37 °C incubator to observe the depth of the purple circle formed on the glass tube. The stained cells were solubilized in 95% ethanol. The absorbance of samples at 590 nm was determined using the Spectramax (Molecular Devices, Sunnyvale, CA, USA).

### 2.6. Swimming Motility Assay

The swimming motility assay of *Xoo* strains was performed on semi-solid medium plates as previously described [36]. *Xoo* strains were grown in NB medium overnight at 28 °C, and the OD_600_ was adjusted to 0.3. Then, 2 μL *Xoo* bacteria were inoculated in the center of the 0.3% semi-solid medium plates (1 g/L yeast extract, 10 g/L sucrose, 5 g/L bacto peptones, and 3 g/L agar) by pipetting. Tested plates were incubated at 28 °C for 3 days. The diameter of circular zones was measured and evaluated.

### 2.7. Growth Measurement and Virulence Assay of Xoo

*Xoo* strains were cultivated in NB medium at 28 °C for 12 h and the OD_600_ was adjusted to 0.05. Then, the samples were inoculated into fresh NB medium for shaken culture at 28 °C for 14 h. The bacterial OD_600_ values were evaluated every 2 h. As previous studies described, pathogenicity investigations of *Xoo* were accomplished in the glasshouse at a temperature of 25–28 °C. Briefly, *Xoo* strains were cultured in NB medium overnight with appropriate antibiotics and collected by centrifugation. The collected cells were resuspended with distilled water and the OD_600_ was adjusted to 0.3. Bacterial suspensions were pressure-infiltrated into the leaves of susceptible rice IR24. The water-soaking regions caused by *Xoo* were quantified using ImageJ software 3 days after infiltration. The OD_600_ of the suspensions was adjusted to 0.6 and inoculated in the rice IR24 by the leaf-clipping method. Disease lesion lengths were observed to evaluate the virulence of *Xoo* strains 14 days after inoculation. There were three independent replications of these experiments.

### 2.8. RNA-Seq and Real-Time Quantitative RT-PCR (qRT-PCR) Analysis

The PΔ*minC*, 8–24, and wild-type PXO99^A^ strains were grown in NA medium and adjusted to OD_600_ = 1.0. The total bacterial RNA was extracted using the EasyPure RNA Kit (TransGen, Beijing, China) and reverse-transcribed to cDNA by the cDNA synthesis kit (Takara, Dalian, China) as per the manufacturer’s protocols. Personalbio (Shanghai, China) accomplished RNA-seq analysis using the Illumina Hiseq platform. The DEGs in *minC* mutant strains were evaluated based on per million reads mapped (FPKM) values and illustrated using the TB tools software as heatmaps [37]. The *Xoo* PΔ*minC*, PΔ*minD*, PΔ*minCDE*, and PXO99^A^ strains were re-suspended with XOM3 medium and shaken cultured at 28 °C for 12 h. Personalbio (Personalbio, Shanghai, China) evaluated RNA-seq on the Illumina Hiseq platform. The expression of *Xoo* genes was analyzed by qRT-PCR employing the ABI 7500 software and SYBR Green I Mix (TransGen, Beijing, China). cDNAs were amplified using the specific primers (Appendix A). The *Xoo **rpoD* and *gyrB* genes were used to normalize the qRT-PCR results, and the 2^−ΔΔCT^ method was used to calculate the gene expression, as previously described. The GOseq R package analyzed DEGs’ Gene Ontology (GO) enrichment. GO terms with a *p*-value of 0.05 were considered significantly enriched. As previously described, we analyzed the Kyoto Encyclopedia of Genes and Genomes (KEGG).

### 2.9. Western Blotting Analysis

The protein expression vectors pH1-*hrpG*::FLAG, pH3-*hrpX*::FLAG, and pH3-*hrpB1*::FLAG were constructed in our previous study [6], then were electroporated into the *Xoo* PXO99^A^, PΔ*minC*, PΔ*minD*, and PΔ*minCDE*, respectively. Overnight, *Xoo* strains were grown in NB medium at 28 °C and collected by centrifugation. Bacterial cells were rinsed with sterile water and resuspended at an OD_600_ of 2.0 in a type III-inducing XOM3 medium. These XOM3 suspensions were incubated in the shaken culture at 28 °C for 12 h. Protein samples were extracted from XOM3 suspension and separated by 10% SDS-PAGE. The proteins were then transferred to a PVDF membrane for immunoblotting using the Flag tag and anti-mouse IgG antibody (TransGen, Beijing, China). The membrane was visualized with the EasySee Western Kit (TransGen, Beijing, China). We used the *E*. *coli* RNA polymerase subunit (RNAP) antibody as the loading control.

### 2.10. GUS Assays

The promoter-probe vectors pHG2-*hrpG*, pHG2-*hrpX*, pHG2-*hrpF*, and pHG3-*hrpB1* were constructed in our previous studies [6,33,36]. These plasmids were transferred into the mutants PΔ*minC*, PΔ*minD*, PΔ*minCDE*, and PXO99^A^ by electroporation. The reporter strains were grown in NA medium with appropriate antibiotics at 28 °C overnight. Then, the *Xoo* cells were collected and cultured in the XOM3 medium for *hrp* induction. The β-glucuronidase (GUS) activities were detected and calculated as previously described [6].

### 2.11. Southern Blotting Analysis

The deletion mutants PΔ*minC*, PΔ*minD*, PΔ*minCDE*, and transposon mutants were further confirmed by Southern blotting analysis according to our previous operation [36]. Total genomic DNA of the Min mutant strains was extracted using the Bacteria Genomic DNA Kit (TransGen, Beijing, China) as the manufacturer recommended. DNA samples of *Xoo* were digested aseptically for 6 h at 37 °C with the restriction enzyme *Bam*HI (Takara Bio, Kusatsu, Japan). Separation and transfer to Hybond N+ nylon membrane using electrophoresis were carried out as described previously [36]. The digoxigenin (DIG)-labeled *minC*, *minD*, and *minCDE* probes and the DIG Easy hybridization buffer (Roche, Sweden) were used for membrane hybridization. The membrane was incubated by detection buffer (Roche) and detected using a digital camera.

### 2.12. Statistical Analysis

All experiments were replicated at least three times independently. The statistical software SPSS v24.0 (SPSS Inc., Chicago, IL, USA) was used to analyze the data, before proceeding with Duncan’s test.

## 3. Results

### 3.1. Min System Participates in Negative Regulation of T3SS Expression

To identify novel T3SS regulators in *Xoo*, we selected two representative *hrp* genes, *hrpF* and *hrpB1,* to construct the plasmid-borne reporters pHG2-*hrpF* and pHG3-*hrpB1*, which contain the *hrpF* and *hrpB1* promoter-*uidA* transcriptional fusion, respectively. The *hrpF* gene was speculated to encode a translocator that transports the T3SE proteins into the host cells [38,39]. The *hrpB1* gene is the first one in the *hrpB* operon of the *Xoo hrp* cluster, the promoter region of which contains a PIP box region that has been demonstrated to bind and activate by HrpX [5]. We screened two mutant 8–24 and 24–46 from ten thousand Tn5 transposon mutants, and found that 8–24 with increased *hrpF* promoter-driven GUS activity in XOM3, a *hrp*-inducing medium (Appendix A), had a Tn5 transposon insertion at 541 bp position of the *minC* gene (Figure 1A), as well as 24–46 with enhanced *hrpB1* promoter-driven GUS activity (Figure 1E), had a Tn5 transposon insertion at 411 bp position of the *minD* gene (Figure 1A). In *Xoo*, the *minC* and *minD* genes along with the *minC* gene are located in an operon, which is highly conserved in xanthomonads such as *Xoo*, *Xcc* and *Xcci* (Appendix A). The *minCDE* genes encode Min system proteins that have been demonstrated to prevent the formation of septa at cell poles by inhibiting the Z-ring, ensuring that bacterial cell division occurs in the middle of cell, not at cell poles in *E*. *coli*, *Bacillus subtilis*, and *Pseudomonas aeruginosa* [18,19,40]. Our PCR analysis, based on the cDNA and genomic DNA of the wild-type PXO99^A^, showed that *minC*, *minD*, and *minE,* with the other two genes, *PXO_04462* and *PXO_04466*, are located in a transcription unit (operon) (Appendix A). *PXO_04462* encodes a putative Gcn5-related N-acetyltransferase (GNAT)-family protein that includes a large number of members among eukaryotes and prokaryotes [41].

To confirm the increase in *hrpB1* and *hrpF* expression in the transposon mutants, we constructed the *minC* deletion mutant PΔ*minC*, the *minD* deletion mutant PΔ*minD*, and the triple mutant PΔ*minCDE* containing the deletion of *minC*, *minD*, and *minE* in the background of *Xoo* wild-type PXO99^A^ using the SacB-based markerless knockout technique. The Southern blotting was performed in the mutant strains to prove the deletions (Appendix A). Similar to the transposon mutant 8–24, PΔ*minC* exhibited a significant increase in GUS activity of *hrpB1* and *hrpF* promoters, *hrpB1* mRNA levels, and HrpB1 protein expression levels compared with the wild-type PXO99^A^ in XOM3 (Figure 1B–D and Appendix A). The enhanced *hrpB1* and *hrpF* expression of PΔ*minC* could be fully restored to the wild-type levels in CPΔ*minC*, a complementary strain of PΔ*minC*, which carries a functional *minC* gene expressed by its native promoter in a low copy number plasmid pML123 (Figure 1B,C and Appendix A). However, the expression of *minC in trans* in 8–24 could partially restore *hrpB1* expression to wild-type levels (Figure 1B). Similarly, like the transposon mutant 24–46, an increase in *hrpB1* and *hrpF* promoter-driven GUS activity, *hrpB1* mRNA levels, and HrpB1 protein expression levels was observed in PΔ*minD* and PΔ*minCDE* compared with that in the wild-type PXO99^A^ (Figure 1E–G and Appendix A). These results indicate that the Min system participates in the negative regulation of T3SS expression.

### 3.2. Min System Inhibits T3SS Expression through the HrpG–HrpX Regulatory Pathway

To further determine whether the Min system regulates *hrpB1* and *hrpF* expression through the key *hrp* regulator HrpG and HrpX, we analyzed the *hrpG* and *hrpX* expression in *minC* and *minD* mutants and the triple mutant PΔ*minCDE*. We measured the GUS activity of the wild-type PXO99^A^, PΔ*minC*, PΔ*minD*, and PΔ*minCDE* carrying the reporters pHG2-*hrpG* and pHG2-*hrpX*, which contain transcriptional fusions of *hrpG* and *hrpX* promoters with the *uidA* gene, respectively. A significant increase in GUS activity of *hrpG* and *hrpX* promoter was observed in PΔ*minC*, PΔ*minD*, and PΔ*minCDE* in comparison to that in wild-type (Figure 2A,B). The enhanced *hrpG* and *hrpX* promoter-driven GUS activity of PΔ*minC* could be restored to the wild-type levels in CPΔ*minC*. In addition, the GUS activity of PΔ*minC*, PΔ*minD*, and PΔ*minCDE* carrying the reporter pHG3-*hrpG*-post that contains a post-transcriptional fusion of *hrpG* with the *uidA* gene, was measured. Similarly, the mutants PΔ*minC*, PΔ*minD*, and PΔ*minCDE* exhibited a dramatic increase in *hrpG* expression-driven GUS activity (Figure 2C). We next investigated the HrpG and HrpX protein expression in the mutants. The Western blotting assays showed that the HrpG and HrpX expression levels were significantly enhanced in *minC* mutants 8–24 and PΔ*minC*, *minD* mutants 24–46 and PΔ*minD*, as well as the triple mutant PΔ*minCDE* compared with that in the wild-type PXO99^A^ (Figure 2D,E). These results suggest that the Min system negatively regulates T3SS expression through the HrpG–HrpX regulatory pathway.

### 3.3. Min System Is Involved in Positive Regulation of Two Key Virulence Regulators RpfG and Clp

To further explore whether the known key virulence regulators, including the quorum-sensing system RpfF/RpfC/RpfG, two *hrpG* positive regulators Trh and XrvA, the transcriptional regulator Clp, and the *hrpX* rather than *hrpG* positive regulator Zur, were involved in the MinCDE-T3SS regulatory pathway, we first analyzed the hrpG promoter activity in the mutants defective in the genes encoding the regulator mentioned above. The quantitative GUS assays indicated that the *hrpG* expression was significantly enhanced in the quorum-sensing mutants PΔ*rpfF*, PΔ*rpfC*, and PΔ*rpfG*, and the clp mutant PΔ*clp* than that in the wild-type PXO99^A^ (Figure 3A). However, the hrpG promoter-driven GUS activity was lower in the *trh* and *xrvA* mutants PΔ*trh* and PΔ*xrvA*, and no differences in the *hrpG* promoter-driven GUS activity were observed between the *zur* mutant PΔ*zur* and the wild-type PXO99^A^ (Figure 3A), which is in agreement with the previous studies [14,42]. Similar results were obtained by the Western blotting assays in which the HrpG expression levels were dramatically higher in PΔ*rpfF*, PΔ*rpfC*, PΔ*rpfG*, and PΔclp than that in the wild-type PXO99^A^, suggesting that RpfF/RpfC/RpfG and Clp functions as a *hrpG* negative regulator. We next investigated the mRNA levels of *rpfF*/*rpfC*/*rpfG*, *clp*, *trh*, and *xrvA* in the Min mutants by qRT-PCR. The results showed that the mRNA levels of *rpfG* were significantly reduced in PΔ*minC*, PΔ*minD*, and PΔ*minCDE*, but the mRNA levels of *clp* were lower in PΔ*minC* and PΔ*minD*, and not in PΔ*minCDE*, compared with the wild-type (Figure 3C). However, the *rpfF*, *rpfC*, and *trh* mRNA levels in the Min mutants were almost the same as that in the wild-type. From these results, we speculate that RpfG and Clp might be involved in T3SS regulation by the Min system in *Xoo*.

### 3.4. Deficiency of the Min System Causes Aberrant Cell Morphology and Division

It has been reported that cells with Min system deficiency fail to prevent the Z-ring from localizing to the cell poles and have aberrant cell division resulting in forming filamentous cells, minicells, or branching [27,30]. We investigated the cell size and shape of the Min mutants defective in *minC*, *minD*, and *minCDE* by transmission electron microscopy (TEM), scanning electron microscope (SEM), and fluorescent microscope (FM). The TEM observation showed that the cell elongation and asymmetric division of 8–24, PΔ*minC*, PΔ*minD*, and PΔ*minCDE* were evident when compared to the wild-type PXO99^A^, which are normal rod-shaped cells (Figure 4A). PΔ*minC* exhibited the classic Min-defective cell phenotypes: minicells and short filamentation observed in other bacteria such as *Xcci* with the *minC* deletion [30], whereas the complementary strain CPΔ*minC* looked normal, like the wild-type (Figure 4B). Although the short filaments were observed in PΔ*minD* and PΔ*minCDE*, the occurrence frequency of short filamentations in the *minC* mutants 8–24 and PΔ*minC* was higher than that in PΔ*minD* and PΔ*minCDE*. Similar phenotypes of minicells and short filamentations were obtained in 8–24, PΔ*minC*, PΔ*minD*, and PΔ*minCDE* carrying a highly expressed green fluorescent protein (GFP) by FM (Figure 4A), indicating that the alterations (short filamentations and minicells) are typical in Min mutants.

To define the length distribution of the Min mutants, we divided the cell populations into four categories based on their cell body lengths: <0.5 µm, minicells; 0.5–1 µm; 1–2 µm; and >2 µm. The standard deviation for each population was obtained after averaging under SEM conditions. The wild-type PXO99^A^ cells had a mean length of 1.43 ± 0.28 µm (*n* = 265) and did not form minicells. The *minC* insertion mutant 8–24 had a mean length of 1.79 ± 1.26 μm (*n* = 259). The shortest minicell of 8–24 was 0.176 µm, and the frequency of minicells was about 7.72%, whereas the most extended cell was 9.135 µm, and the proportion of filamentous cells was about 33.59% (Appendix A and Table 1). The *minC* deletion mutant PΔ*minC* possesses a mean length of 1.32 ± 0.633 µm (*n* = 268), with filamentous cells recording for 11.94% and minicells accounting for 2.99% (Appendix A and Table 1). The recovered strain CPΔ*minC* possesses a mean length of 1.58 ± 0.50 µm (*n* = 264) without the minicells, indicating similar morphology (cell shape and cell length variation) to the wild-type PXO99A. Minicells were almost absent in PΔ*minD* (*n* = 257) and PΔ*minCDE* (*n* = 258), while the ratios of cells length than 2 µm (18.29% and 13.18%) in PΔ*minD* (*n* = 257) and PΔ*minCDE* (*n* = 258) were significantly longer than that in the PXO99^A^ (Appendix A and Table 1). These results indicated that mutation of *min* genes, especially the *minC* gene, causes aberrant morphology and asymmetric division of *Xoo* cells.

### 3.5. Effect of Min System on Bacterial Virulence, Motility, and Biofilm Formation

In *Xoo*, the T3SS is essential for bacterial pathogenicity on susceptible host rice and triggering HR on nonhost. To verify whether the deletion of *min* genes affects *Xoo* virulence, we inoculated the Min mutants and relative complementary strains on IR24, a susceptible rice variety, by the leaf-clipping method. The result showed that all mutants could cause the water-soaked lesions on IR24 (Appendix A), whereas the *minC* insertion mutant 8–24 exhibited a significant decrease in lesion length on IR24 when compared to the wild-type PXO99^A^, and the deletion mutants PΔ*minC*, PΔ*minD*, and PΔ*minCDE* displayed a weaker reduction in virulence on IR24 (Figure 5A). The corresponding complementary strains CPΔ*minD* and CPΔ*minCDE*, in which the *minD* and *minCDE* genes were expressed in trans, could be retained the wild-type ability to cause lesion length on IR24, indicating that the Min system is required for *Xoo* full virulence on host rice. The inoculation assays on tobacco indicated that the absence of the Min system did not affect the capacity of *Xoo* to trigger HR on nonhost tobacco (Appendix A).

It has been shown that Min system proteins prevent the septa formation at the cell pole by inhibiting the Z-ring [19,30]. Taking into account that the swimming motility of *Xoo* is dependent on a polar flagellum, we explored the role of the Min system in *Xoo* swimming motility, and conducted the swimming motility assays in which the Min mutants were inoculated on the semi-solid NA medium with 0.15% agar. Similar to the transposon mutant 8–24, the *minC* deletion mutant PΔ*minC* did not exhibit any significant swimming motility, but the *minD* mutants 24–46 and PΔ*minD*, as well as the triple mutant PΔ*minCDE* showed slightly reduced swimming motility compared to the wild-type PXO99^A^ (Figure 5B). The complementary strain CPΔ*minD* nearly reverted swimming motility to wild-type levels (Appendix A). Because swimming motility has been indicated to be related to biofilm formation, we sought to characterize the biofilm formation of the Min mutants. The crystal violet staining assays showed that the *minC* mutants 8–24 and PΔ*minC* produced more significant biofilm than the wild-type PXO99^A^, and the triple mutant PΔ*minCDE* had slightly larger biofilm than PXO99^A^; however, the *minD* mutants 24–46 and PΔ*minD* produced no differences in biofilm with the wild-type PXO99^A^ (Figure 5C). These results demonstrate that merely the mutation of *minC* significantly affects the swimming motility and biofilm formation of *Xoo*, indicating that each of the min genes plays a different role in swimming motility and biofilm formation.

### 3.6. The Transcriptome Profiling Reveals Virulence-Relevant Genes Affected by MinC

To further explore the function of *minC* in *Xoo* virulence, we analyzed the differentially expressed genes (DEGs) in the *minC* mutants 8–24 and PΔ*minC* compared with the wild-type PXO99^A^. Based on standards of log2 (fold change) ≥1 or ≤−1 (*p* value < 0.05), we screened 198 and 106 DEGs in PΔ*minC* and 8–24, respectively (Appendix A). Heatmap analysis displayed the base mean DEGs in wild-type PXO99^A^, PΔ*minC*, and 8–24 strains (Figure 6A,B). Three genes (*PXO_04154*, *PXO_04552*, and *PXO_04756*) were significantly down-regulated in 8–24 (Figure 6A). *PXO_04756* was reported to have a role related to cardiolipin synthesis. Furthermore, the expression of 99 genes, including two copies of *clpA* (*PXO_06136* and *PXO_01030*), was significantly up-regulated. ClpA was annotated as an ATP-binding subunit of the Clp protease. In PΔ*minC*, 14 and 184 genes were significantly down-regulated and up-regulated, respectively (Appendix A). Six *hrp* genes, *hrpD6* (*PXO*_*03410*), *hpaA* (*PXO_03408*), *hpaB* (*PXO_03412*), *hrcU* (*PXO_03402*), *hrpD5* (*PXO_03409*) and *hrpE* (*PXO_03411*), and the TCS genes *raxH* (*PXO_04467*) and *raxR* (*PXO_04469*) were all up-regulated in the *minC* mutant PΔ*minC* (Figure 6C). The expression levels of these genes were dramatically higher in 8–24 than in wild-type PXO99^A^. Furthermore, the cytokinesis-related gene *zipA* (*PXO_00742*) was also significantly enriched. In PΔ*minC* and 8–24, Venn diagram analysis revealed that 78 up-regulated DEGs and 2 down-regulated DEGs were overlapped, demonstrating the precision of the RNA-seq.

We investigated DEGs for GO enrichment in PΔ*minC* and 8–24, respectively (Figure 6D and Appendix A). All DEGs were classified into three main categories based on their putative function. The findings revealed that the majority number of DEGs enriched in biological processes in 8–24, with significant enrichment in the homeostatic process (GO:0042592), including three up-regulated DEGs, *copB* (*PXO_03131*), *cutC* (*PXO_01619*), and ferripyoverdine receptor (*PXO_03287*). These results suggest that MinC can regulate DEGs expression by altering the function of the homeostatic process. As with 8–24, PΔ*minC* concentrated a significant proportion of the DEGs in biological processes. The DEGs up-regulated in PΔ*minC* are highly enriched in functions associated with protein maturation (GO: 0051604) and transferase activity involved in the alkyl or aryl transfer (GO: 0016765). The interaction between MinC and transferase activity genes might occur in stress response modulation and differential stability. These genes are possibly associated with the virulence mediated by MinC.

The KEGG pathway would be used to categorize further and study the biological functions of these DEGs. Among these pathways, a *p*-value ≤ 0.05 was necessary for analysis. DEGs are implicated in various pathways. The DEGs influence six critical pathways in strains 8–24 (Appendix A), including cell growth and death, immune disease, and infecting diseases. Similarly, DEGs in PΔ*minC* were considerably more abundant in primary immunodeficiency (ko05340) of immunological disorders (Figure 6E). In PΔ*minC*, the uracil-DNA glycosylase gene (UDG, *PXO_03712*) expression was increased significantly. UDGs exist in different bacteria and possess base activity to excise damaged bases in DNA. Furthermore, they can increase heat resistance. These results suggest the potential stress resist functions of MinC in *Xoo*.

## 4. Discussion

The Min system comprising three proteins, MinC, MinD, and MinE, is conserved among genera of rod-shaped bacteria, and its function in cell division has been well studied in *E*. *coli* and *B*. *subtillis*. However, most studies focused on understanding its role in interaction with other divisome proteins such as FtsZ, whereas other roles in cellular processes including virulence, bacterial motility, and colonization were not explored. In this study, we found that apart from involvement of the *Xoo* Min system in cell division, the Min system also participates in the regulation of T3SS expression, bacterial full virulence, swimming motility, and biofilm formation, suggesting that the function of Min proteins is not strictly confined to cytokinesis; more cellular functions must be elucidated.

Generally, the Min system mutation leads to abnormal morphology such as minicells, short filamentations, and branching in bacteria [17,30]. Our microscopy observations showed that the 8–24, PΔ*minC*, PΔ*minD*, and PΔ*minCDE* mutants exhibited obvious cell elongation and asymmetric division. The aberrant cell division phenotypes including minicells and short filamentations were observed in 8–24, PΔ*minC*, PΔ*minD* and PΔ*minCDE*, especially in the *minC* 8–24 and PΔ*minC* mutans, which are classic Min-defective cell phenotypes, and also similar to cell shapes of the *minC*-defective mutants in *Xcci* 306 and *Helicobacter pylori* [27,30], indicating that the *Xoo* Min system indeed plays a key role in cell division. However, some branching cells comprising less than 20% of total cells were observed in the *minC* mutant of *Xcci* 306 [30], which were not observed in our microscopy assays. Branched cells impair the divisome formation, the nucleoid organization, and the incorporation of peptidoglycans. The phenotype of branching cells reported in the *E*. *coli* Min mutants was dependent on the growth medium used in the experiments [43,44]. Almost no branching cells were observed in the *minC* mutant of *Xcci* when the rich NYG/CB media were used [30]. We speculated that the absence of branching cells in the Min mutants of *Xoo* might be the reason for the nutrient-rich NB medium employed in our assays.

Some studies in pathogenic bacteria have shown that Min proteins are essential for full virulence. The *minD* mutant of the pathogenic enterohemorrhagic *E*. *coli* (EHEC) reduced its adherence to the human epithelial tissues [45]. Both mutations of *minC* and *minD* in *Neisseria gonorrhea*, a sexually-transmitted bacterium, reduced its ability to adhere to and invade urethral epithelial cells, but did not alter its potential to produce other virulence factors [46]. Our results showed that the *minC* insertion mutant 8–24 exhibited an attenuated virulence in rice, whereas the deletion mutants PΔ*minC*, PΔ*minD*, and PΔ*minCDE* displayed a weaker reduction in virulence. We speculate that the different phenotypic effects on virulence between 8–24 and PΔ*minC* could be related to the mutation sites in the *minC* gene in these two mutants. We deleted the middle open reading fragment of *minC* in PΔ*minC*, but the Tn5 transposon was inserted in the 3’-terminal of *minC* in 8–24. This suggests that the C-terminal domain of MinC is important for the function of MinC in bacterial virulence. Similarly, a current study in *Xoo* PXO99^A^ has shown that MinD was significantly downregulated during its interaction with host rice IR24, and the average lesion lengths caused by the *minD* mutant were significantly shorter than those caused by the wild-type PXO99^A^ [31]. Taken together, these results indicate that MinC and MinD are essential for *Xoo* full virulence in susceptible host rice.

Our swimming motility assays showed that the mutants with inactivation of *minC*, *minD*, or *minCDE* showed reduced swimming motility as compared to the wild-type PXO99^A^. However, the *minC* mutants 8–24 and PΔ*minC* nearly lost swimming motility, indicating that MinC plays a critical role in swimming motility. This result is in agreement with some studies in *Proteus mirabilis* and *H. pylori* [26,27]. Both *minC* mutants in these two bacteria exhibited reduced swarming motility. It has been determined that alteration in cell morphology might affect motility. Although the mutations of *minC*, *minD*, and *minCDE* resulted in elongation in cell lengths, a *minC* mutation alone was found to lose swimming motility. Therefore, we speculate that asymmetric division may affect bacterial motility whereas, more to the point, some underlying mechanisms or connections between MinC and flagellar biosynthesis are essential for swimming motility. Current studies have indicated some relation between the Min system and flagella regulators, such as FlhG and FlhDC, a master regulator for flagellar synthesis [17,26,47]. These findings also indicate that *Xoo* is an ideal model bacterium to study the role of cell division proteins in motility function.

In this study, we discovered that the Min system is extensively conserved in seven species of the genus *Xanthomonas*, and the *minCDE* gene cluster co-transcribed with the flanking genes *PXO_04462* and *PXO_04466*. A similar study has been observed in pathogenic *N*. *gonorrhea*, in which the *minCDE* gene cluster is transcribed with *oxyR*, which encodes a redox-response transcriptional regulator (LysR-NodD family) that can directly bind the promoter regions of some catalase genes such as *katA* [28]. The mutation of *N*. *gonorrhea oxyR* led to defective cell division and enhanced *minD* expression [28]. The *Xoo PXO_04462* gene encodes a putative GNAT-family protein. The C-terminal domain of the GNAT family contains an acetyl-CoA binding fold that transfers the acetyl group from acetyl-CoA to a variety of N-terminal amino groups. The mutation of the GNAT gene in *Dickeya zeae* MS2 has been shown to decrease virulence in potatoes [48]. We have constructed the deletion mutants of *PXO*_*04462* and *PXO*_*04466*. Whether these two genes are involved in cell division inhibition and expression of *minCDE* genes needs to be examined further. Moreover, we found that *raxR*-*raxH*, a pair of genes associated with a two-component system directly orthologous to *Pseudomonas colS*-*colR* [49,50], was located upstream of the *minCDE* operon in *Xoo*. Our RNA-seq data showed that the expression of *raxH* and *raxR* was significantly higher in the *minC* mutants than that in the wild-type. It has been determined that, in response to Zn^2+^ stress, RaxH-RaxR regulates the *arnT*-*lpxT*-*eptA* gene cluster to participate in lipid A remodeling enzyme synthesis [49,51]. Therefore, we hypothesized that the *Xoo* Min system might be involved in other cellular processes associated with stress response.

Our study demonstrated that the Min system inhibited the *hrp* genes (*hrpB1* and *hrpF*) expression through HrpG and HrpX in XOM3. To our knowledge, this is the first report about the Min system participating in the regulation of T3SS expression in *Xoo*. This finding is further confirmed by the RNA-seq data, by which we found that *hrpD6* (*PXO*_*03410*), *hpaA* (*PXO_03408*), *hpaB* (*PXO_03412*), *hrcU* (*PXO_03402*), *hrpD5* (*PXO_03409*), and *hrpE* (*PXO_03411*) were induced in the *minC* mutants 8–24 and P∆*minC*. These results are consistent with a current finding that MinD was significantly downregulated during early interaction of *Xoo* with host rice IR24 [31]. As the reduced MinD expression causes increased expression of *hrp* genes, it is logical for inducible expression of *hrp* genes in the early stage of interaction with host rice. Our results showed that the mutations of *minC*, *minD*, or *minCDE* caused an increase in *hrp* genes (*hrpF* and *hrpB1*), but the mutants PΔ*minC*, PΔ*minD*, and PΔ*minCDE* displayed a weaker reduction in virulence. We speculate that high expression of *hrp* genes does not necessarily cause an increase in bacterial virulence on the host plant. For example, in our previous study, a *metB* mutant of *Xoo* PXO99^A^ exhibited the enhanced *hrpG* expression in XOM3, but showed impaired virulence in host rice, as the *metB* gene is the EPS and LPS synthesis-related gene [52]. Our results showed that *rpfG* and *clp* were down-regulated in the *minC* and *minD* mutants, whereas *hrpG* was up-regulated in the *rpfG* and *clp* mutants. RpfG is a response regulator of the two-component system RpfG/RpfC with the capacity of degrading c-di-GMP, and Clp is a homologue of cyclic AMP receptor protein (CRP) with the ability to bind c-di-GMP [53,54]. Therefore, DSF and c-di-GMP (or cAMP) signal pathways were speculated to participate in T3SS expression regulated by the Min system in *Xoo*. Our RNA-seq data also indicated that MinC regulates the expression of two copies of *clpA*. ClpA, a Clp protease, has been demonstrated to be a virulence factor in *Xoo* and protect the cytoplasm against the detrimental effects of stressful conditions imposed by host defense mechanisms and environmental events [55]. Taken together, we speculate that negative regulation of T3SS expression by the Min system in *Xoo* is complex, and that a combination is involved in multiple signaling pathways.

## 5. Conclusions

In this study, we identified the *Xoo* Min system (MinC, MinD, and MinE) functioning as a negative regulator for T3SS expression through the key *hrp* regulators HrpG and HrpX. The mutations of *minC*, *minD*, and *minCDE* resulted in cell elongation and asymmetric division; meanwhile, mutation of *minC* in *Xoo* resulted in significantly impaired virulence in host rice, swimming motility, and enhanced biofilm formation. Our transcriptome profiling also indicated that some virulence genes were differentially expressed in the *minC* mutants. To our knowledge, this is the first report about the Min system participating in the regulation of T3SS expression. It provides some evidence for the complex T3SS regulatory networks and sheds light on the understanding of *Xoo* virulence mechanisms.

## Figures and Tables

**Figure 1 microorganisms-10-01549-f001:**
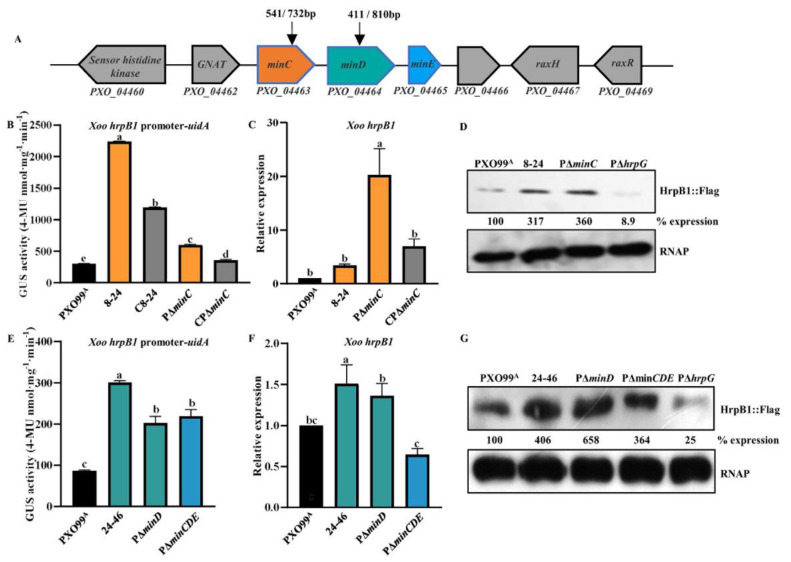
The *Xoo* Min system negatively regulates *hrpB1* expression. (**A**) Genomic location of Min system on the *Xoo* PXO99^A^ chromosome. Arrows indicate the Tn5 transposon insertion sites in the nucleotides 541 of *minC* and the nucleotides 411 of *minD*, respectively. (**B**) The *hrpB1* promoter-driven GUS activity of *Xoo* wild-type PXO99^A^, 8–24, C8–24, P∆*minC*, and CP∆*minC* in XOM3 at 3 hr post-induction. (**C**) Expression ratios of *hrpB1* in 8–24, P∆*minC*, and CP∆*minC* compared to that in *Xoo* wild-type PXO99^A^ by qRT-PCR. (**D**) The abundance of HrpB1 proteins in PXO99^A^, 8–24, and P∆*minC* by Western blotting. P∆*hrpG* as a negative control of HrpB1 protein expression. (**E**) The *hrpB1* promoter-driven GUS activity of PXO99^A^, 24–46, P∆*minD*, and P∆*minCDE* in XOM3 at 3 hr post-induction. (**F**) Expression ratios of *hrpB1* in 24–46, P∆*minD*, and P∆*minCDE* compared to that in *Xoo* wild-type PXO99^A^ by qRT-PCR. (**G**) The abundance of HrpB1 proteins in PXO99^A^, 24–46, P∆*minD*, and P∆*minCDE* by Western blotting. P∆*hrpG* as a negative control of HrpB1 protein expression. The total protein extracts were analyzed by Western blotting using anti-FLAG antibodies. RNAP, RNA polymerase subunit alpha from *E*. *coli* was used as a loading control. Relative protein abundance was calculated by ImageJ software. Similar results were observed in two independent experiments. As assessed by Duncan’s test, different letters indicate statistically significant differences, and the same letter displays no significant differences (*p* < 0.05) between *Xoo* strains.

**Figure 2 microorganisms-10-01549-f002:**
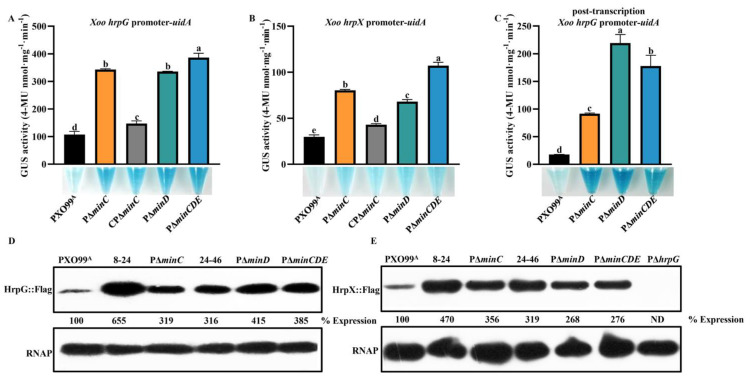
The Min system negatively regulates the expression of *hrpG* and *hrpX*. (**A**) The *hrpG* promoter-driven GUS activity of *Xoo* wild-type PXO99^A^, P∆*minC*, CP∆*minC*, P∆*minD*, and P∆*minCDE* in XOM3 at 3 hr post-induction. (**B**) The *hrpX* promoter-driven GUS activity of PXO99^A^, P∆*minC*, CP∆*minC*, P∆*minD*, and P∆*minCDE* in XOM3 at 3 hr post-induction. (**C**) The *hrpG* expression-driven GUS activity of PXO99^A^, P∆*minC*, P∆*minD*, and P∆*minCDE* harboring a post-transcription *hrpG*::*uidA* fusions in XOM3 at 12 hr post-induction. (**D**) The abundance of HrpG proteins in PXO99^A^, 8–24, P∆*minC*, 24–46, P∆*minD*, and P∆*minCDE* by Western blotting. The data revealed that mutation of Min system increased the HrpG protein levels by more than 3.16-fold. (**E**) The abundance of HrpX proteins in PXO99^A^, 8–24, P∆*minC*, 24–46, P∆*minD*, and P∆*minCDE* by Western blotting. P∆*hrpG* as a negative control of HrpX protein expression. The data revealed that mutation of Min system increased the HrpX protein levels by more than 2.68-fold. The total protein extracts were analyzed by Western blotting using anti-FLAG antibodies. RNAP, RNA polymerase subunit alpha from *E*. *coli* was used as a loading control. Relative protein abundance was calculated by ImageJ software. Similar results were observed in two independent experiments. As assessed by Duncan’s test, different letters indicate statistically significant differences, and the same letter displays no significant differences (*p* < 0.05) between *Xoo* strains.

**Figure 3 microorganisms-10-01549-f003:**
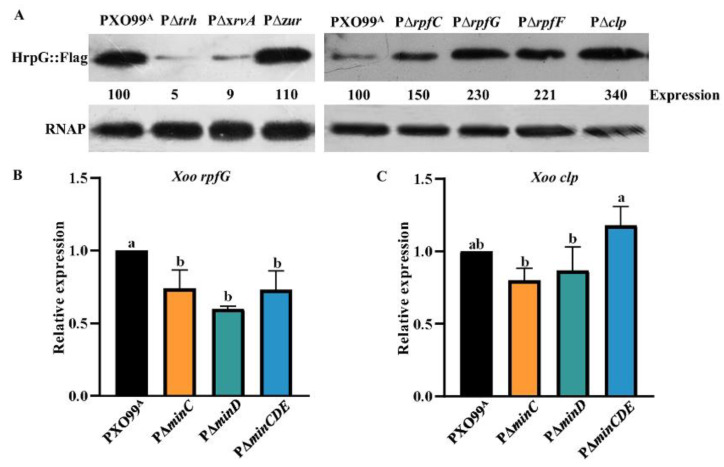
HrpG expression levels in the mutants of *Xoo* virulence regulator genes and the *rpfG* and *clp* expression levels in the Min mutants. (**A**) The abundance of HrpG proteins in PXO99^A^, P∆*trh*, P∆*xrvA*, P∆*zur*, P∆*rpfC*, P∆*rpfG*, P∆*rpfF*, and P∆*clp* by Western blotting. The total protein extracts were analyzed by Western blotting using anti-FLAG antibodies. RNAP, RNA polymerase subunit alpha from *E*. *coli* was used as a loading control. Relative protein abundance was calculated by ImageJ software. Similar results were observed in two independent experiments. (**B**) Expression ratios of *rpfG* in PΔ*minC*, PΔ*minD*, and PΔ*minCDE* compared to that in *Xoo* wild-type PXO99^A^ by qRT-PCR. (**C**) Expression ratios of *clp* in PΔ*minC*, PΔ*minD*, and PΔ*minCDE* compared to that in PXO99^A^ by qRT-PCR. Similar results were observed in more than three independent experiments. As assessed by Duncan’s test, different letters indicate statistically significant differences, and the same letter displays no significant differences (*p* < 0.05) between *Xoo* strains.

**Figure 4 microorganisms-10-01549-f004:**
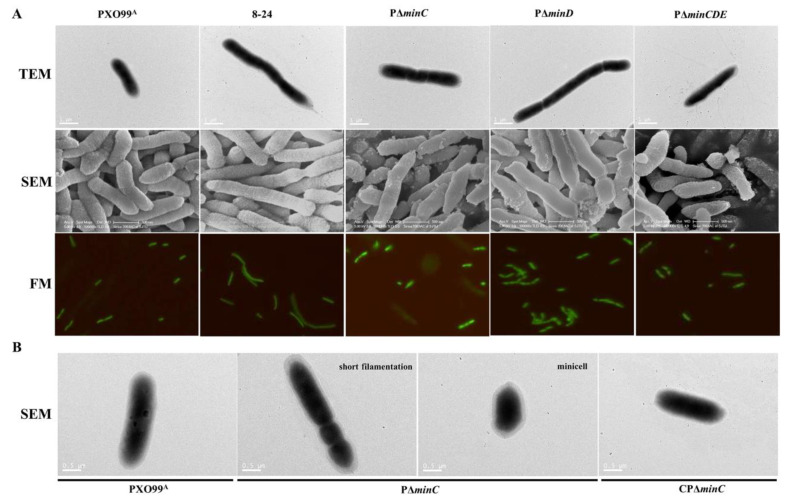
Deficiency of the Min system causes aberrant cell morphology. (**A**) Microscopy images of wild-type PXO99^A^, 8–24, the *minC* mutant PΔ*minC*, the *minD* mutant PΔ*minD*, and the *minCDE* mutant PΔ*minCDE*. Cells were observed by scanning electron microscopy (SEM), transmission electron microscopy (TEM) and fluorescence microscopy (FM). Scale bars: 1, and 0.5 μm. (**B**) Minicells and short filamentations, the classic Min-defective cell phenotypes, were observed in the *minC* mutant PΔ*minC* by SEM. Scale bars: 0.5 μm.

**Figure 5 microorganisms-10-01549-f005:**
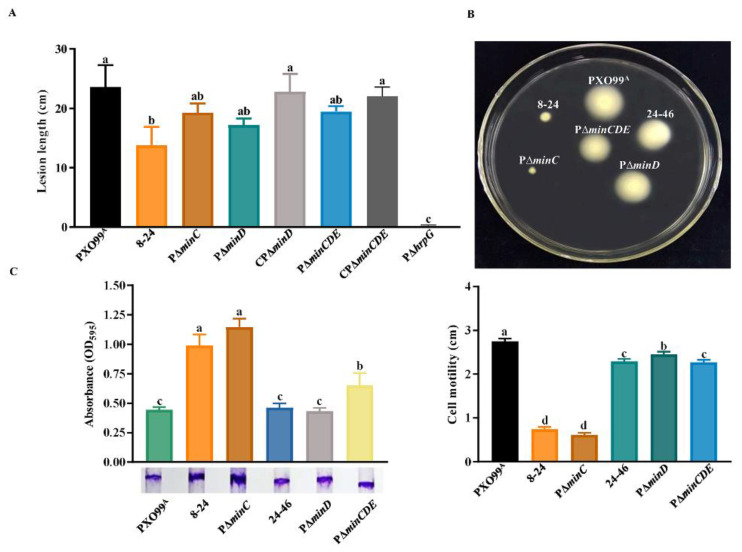
MinC affects *Xoo* virulence, biofilm formation, and swimming motility. (**A**) Lesion lengths of the leaves of IR24 caused by *Xoo* wild-type PXO99^A^, 8–24, PΔ*minC*, PΔ*minD*, CPΔ*minD*, PΔ*minCDE*, CPΔ*minCDE*, and PΔ*hrpG* at 14 days post-inoculation by leaf-clipping. Bacterial suspensions (OD_600_ = 0.6) were inoculated in the leaves of susceptible rice IR24. P∆*hrpG* as a negative control strain without pathogenicity on rice. Similar results were observed in two independent experiments. (**B**) Swimming motility of PXO99^A^, 8–24, PΔ*minC*, 24–46, PΔ*minD*, and CPΔ*minD* on NA medium with 0.15% agar. Swimming zones were measured and evaluated after bacterial growth on the NA plates for 3 days. As assessed by Duncan’s test, different letters indicate statistically significant differences (*p* < 0.05) between *Xoo* strains. (**C**) Biofilm formation of PXO99^A^, 8–24, PΔ*minC*, 24–46, PΔ*minD*, and PΔ*minCDE* on glass test tube surfaces after 3 days of incubation The biofilm formation was visualized by crystal violet staining, then was quantified by measuring the absorbance at 590 nm. The tests were repeated three times. As assessed by Duncan’s test, different letters indicate statistically significant differences (*p* < 0.05) between *Xoo* strains.

**Figure 6 microorganisms-10-01549-f006:**
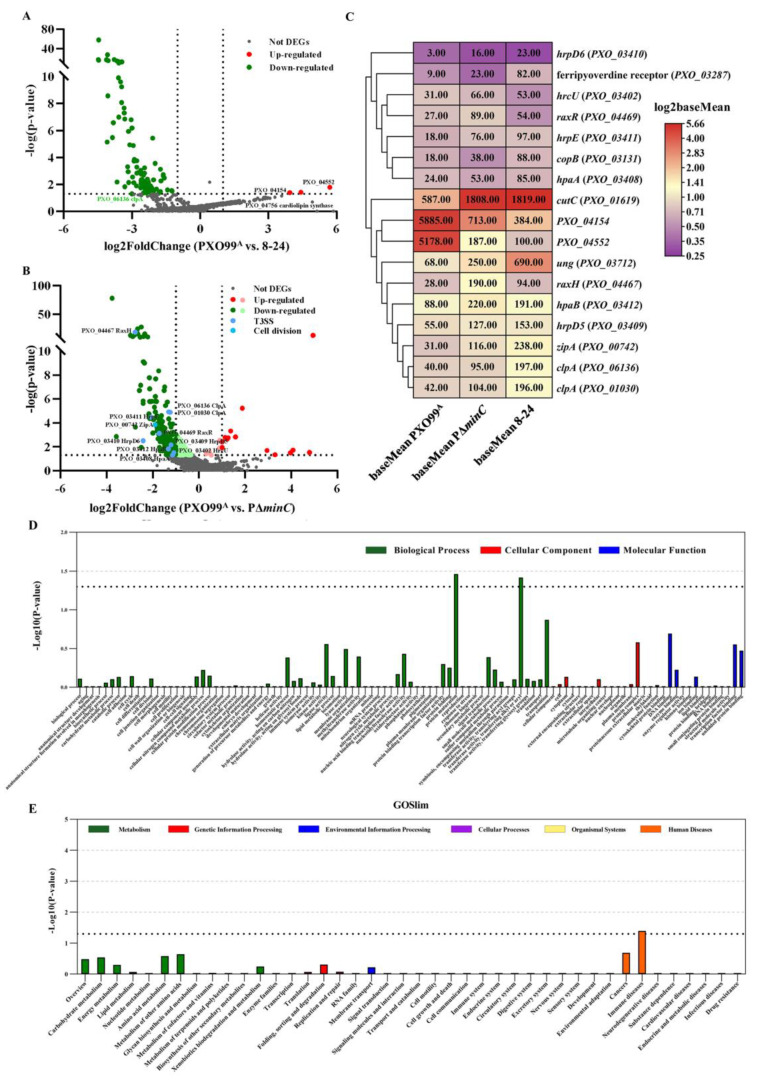
Analysis of the differentially expressed genes (DEGs) in PXO99^A^ versus in PΔ*minC* and 8–24. (**A**) The log_2_Foldchange of DEGs in 8–24 and (**B**) PΔ*minC* was plotted against the *p*-value. Statistically significant differentially expressed genes, with a log_2_Foldchange ≥ 1 or ≤ −1, are depicted as the red and green dots, respectively, and insignificant as grey dots. For each organism, the shade of the color represents the level of gene expression. Dark blue dots represent T3SS-associated genes. Light blue dots represent cell division-associated genes. (**C**) Heatmap of gene expression of the differentially expressed genes (DEGs) in *minC* mutants 8–24 and PΔ*minC*. The color gradient indicates the normalized base mean values of DEGs (high expression (red) and low expression (purple)). (**D**) GO analysis of DEGs in PΔ*minC* mutant. The abscissa axis represents the GO category, and the ordinate axis represents the value of significance (*p* < 0.05). (**E**) KEGG analysis of DEGs in PΔ*minC* mutant. The abscissa axis represents the KEGG pathway, and the ordinate axis represents the value of significance (*p* < 0.05).

**Table 1 microorganisms-10-01549-t001:** Statistical analysis of cell lengths of the Min mutants and the *Xoo* wild-type.

Strain	Minicells Percentage	Filamentous Cells Percentage	Mean Cell Length (µm)	Minimum (µm)	Maximum (µm)
PXO99^A^	0%	4.15%	1.432 ± 0.281	0.791	2.717
8–24	7.72%	33.59%	1.793 ± 1.260	0.176	9.135
PΔ*minC*	2.99%	11.94%	1.318 ± 0.628	0.233	4.630
CPΔ*minC*	0%	17.05%	1.579 ± 0.496	0.582	3.820
PΔ*minD*	0.40%	16.80%	1.521 ± 0.462	0.500	4.155
PΔ*minCDE*	0.80%	13.18%	1.426 ± 0.402	0.405	3.672

## Data Availability

The data that support the findings of this study are available from the corresponding author upon reasonable request.

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
