# Peer review of "The MinCDE Cell Division System Participates in the Regulation of Type III Secretion System (T3SS) Genes, Bacterial Virulence, and Motility in Xanthomonas oryzae pv. oryzae"

_microorganisms, 2022, doi:10.3390/microorganisms10081549_

Round 1

Reviewer 1 Report

This manuscript by Yan et al. shows that Min system (MinC, MinD, and MinE) encoded by minC, minD, and minE genes is involved in negative regulation of hrp genes (hrpB1 and hrpF) in Xoo. The deletion of minC, minD and minCDE resulted in enhanced hrpB1 and hrpF expression, which is dependent on two key hrp regulators HrpG and HrpX. And the mutation of minC in Xoo resulted in significantly impaired virulence in host rice. I think that this paper provides clear supporting results for the conclusion. However, I also find the concerns about somehow contradictory results that mutation of minC increases enhanced hrpB1 and hrpF expression, related to virulence and at the same time, the mutant strain shows impaired virulence. In addition, the enhanced biofilm formation helps Xoo to attach to host cells.

The global transcription factors such as HrpX and HrpG could be involved in many cellular processes. And cell division processes also could affect many. Authors need to explain or summarize the overall effects of mutations in the Min system in terms of pathogenicity.    

Line 406: “all mutants could cause the water-soaked lesions on IR24 (Figure S5A), whereas the minC insertion mutant 8-24 exhibited a significant decrease in lesion length on IR24 when compared to the wild-type PXO99A, whereas the deletion mutants PΔminC, PΔminD, and PΔminCDE displayed a weaker reduce in virulence on IR24 (Figure 5A).” It is not clear that the different phenotypic effects from between 8-24 and the deletion mutants. Explanation or discussion is necessary.

Line 536: also shows similar results

Line 96: “The putative components of the Xoo Min system are encoded by the minC, minD, and minE genes” the sentence is unclear. Rephrasing the sentence is necessary.

Author Response

Response to Reviewer 1

Reviewer 1: This manuscript by Yan et al. shows that Min system (MinC, MinD, and MinE) encoded by minC, minD, and minE genes is involved in negative regulation of hrp genes (hrpB1 and hrpF) in Xoo. The deletion of minC, minD and minCDE resulted in enhanced hrpB1 and hrpF expression, which is dependent on two key hrp regulators HrpG and HrpX. And the mutation of minC in Xoo resulted in significantly impaired virulence in host rice. I think that this paper provides clear supporting results for the conclusion. However, I also find the concerns about somehow contradictory results that mutation of minC increases enhanced hrpB1 and hrpF expression, related to virulence and at the same time, the mutant strain shows impaired virulence. In addition, the enhanced biofilm formation helps Xoo to attach to host cells. The global transcription factors such as HrpX and HrpG could be involved in many cellular processes. And cell division processes also could affect many. Authors need to explain or summarize the overall effects of mutations in the Min system in terms of pathogenicity.

Author's Notes to Reviewer 1:

The point-by point response from authors to the Reviewer is shown in blue.

  1. I also find the concerns about somehow contradictory results that mutation of minC increases enhanced hrpB1 and hrpF expression, related to virulence and at the same time, the mutant strain shows impaired virulence. Author's Notes:The authors previously believed, as did the reviewer, that high expression of hrp genes should lead to increased virulence of pathogens, but this has now changed as more and more evidence (our published and unpublished data) suggests that this logic is not necessarily true. For example, in our previous study, a metB mutant of Xoo PXO99A exhibited the enhanced hrpG expression, but showed impaired virulence in host rice as the metB gene is the EPS and LPS synthesis related genes (Wang, M.; Liu, Z.; Dong, Q.; Zou, L.; Chen, g. Identification of genes regulating hrpG expression in Xanthomonas oryzae pv. oryzae. Acta Phytopathologica Sinica 2015, 45, 130-138.) We're still figuring out the mechanism, and we think there might be 1. Xoo-rice interaction may be a dynamic process. The mutation of minCDE system may lead to the increased expression of hrp genes in the early stage of interaction, and the regulation pattern might be changed in the late stage of interaction. 2. The increased expression of hrp gene may play a role in advancing the interaction between Xoo and rice. 3. Increased hrp gene expression (or increased T3SE transport efficiency, which may play a role in the initial stage of interaction, and later pathogen expansion may be related to other virulence factors such as extracellular cell wall degrading enzymes). We also added the above thoughts in the discussion section.
  2. In addition, the enhanced biofilm formation helps Xoo to attach to host cells. Author's Notes:In Xoo, some evidences (our published data) have shown that increased biofilm does not necessarily enhance the virulence of the pathogen. For example, we have shown that the Xoo PXO99A pilN mutant produces more biofilm but its virulence on host rice is significantly reduced. See reference [36]. We added the corresponding discussion for this result in the Discussion section.
  3. The global transcription factors such as HrpX and HrpG could be involved in many cellular processes. And cell division processes also could affect many. Authors need to explain or summarize the overall effects of mutations in the Min system in terms of pathogenicity. Author's Notes:In terms of this cognition, the authors agreed with the opinions of the reviewer. Previous studies have shown that HrpG and HrpX regulate not only hrp gene expression, but also other virulence related phenotypes. Because the MinCDE system affects hrp gene expression, other phenotypes may be affected through the HrpG and HrpX pathways. These reflections and explanations have been supplemented in the corresponding discussion section.
  4. Line 406: “all mutants could cause the water-soaked lesions on IR24 (Figure S5A), whereas the minC insertion mutant 8-24 exhibited a significant decrease in lesion length on IR24 when compared to the wild-type PXO99A, whereas the deletion mutants PΔminC, PΔminD, and PΔminCDE displayed a weaker reduce in virulence on IR24 (Figure 5A).” It is not clear that the different phenotypic effects from between 8-24 and the deletion mutants. Explanation or discussion is necessary. Line 536: also shows similar results. Author's Notes:We added the corresponding discussion for this result in the Discussion section. We think that the different phenotypes result from different mutation sites in the minC gene in the insertion mutant 8-24 and the deletion mutant PΔminC. We deleted the middle ORF segment of minC in PΔminC, but the Tn5 transposon was inserted in the 3’-terminal of minC in 8-24. This suggests that the C-terminal domain of MinC is important for the function of MinC in bacterial virulence.
  5. Line 96: “The putative components of the Xoo Min system are encoded by the minC, minD, and minE genes” the sentence is unclear. Rephrasing the sentence is necessary. The putative components of the Xoo Min system are encoded by the minC, minD, and minE genes. Author's Notes:we re-wrote this sentence as “The Xoo PXO99A Min system is composed of MinC, MinD and MinE proteins that are encoded by the minC (PXO_04463), minD (PXO_04464), and minE (PXO_04465) genes, respectively. “

Reviewer 2 Report

The manuscript is well written and well organized with sufficient results to conclude the proposed mechanism. Hence i accept the manuscript in its current form.

Author Response

Response to Reviewer 2

The manuscript is well written and well organized with sufficient results to conclude the proposed mechanism. Hence i accept the manuscript in its current form. Author's Notes: Thank you very much for the reviewer's affirmation. We double checked the manuscript, corrected some spelling mistakes, and revised or added some sentences according to the opinions of other reviewers.

Reviewer 3 Report

 In this work the authors studied Xanthomonas oryzae pv. oryzae (Xoo) which causes bacterial leaf blight (BLB), one of the most serious rice bacterial diseases. It was found that type III secretion system (T3SS) of Xoo is encoded by the hypersensitive response and pathogenicity (hrp) genes and is crucial for its pathogenic action on rice plants. The authors then identified the Min system (MinC, MinD, and MinE), a negative regulatory system for bacterial cell division encoded by minC, minD, and minE genes, which is involved in negative regulation of hrp genes (hrpB1 and hrpF) in Xoo. The deletion of  minC, minD and minCDE caused an enhanced hrpB1 and hrpF expression and produced mutants that exhibited elongated cell lengths and the classic Min system-defective cell morphology including minicells and short filamentations and consequently caused significantly impaired virulence in host rice.

The topic of study is relevant to the field because in a multistep study the authors analyzed the role of Min system in the regulation of T3SS expression and the study resulted in better knowledge of Xoo virulence mechanism which in the future could help to find ways to prevent or reduce this disease in rice.

The study was well designed and the resulting data is presented in a good and coherent manner.

The conclusions are appropriate and based on the obtained data.

The references are relevant and sufficient.

Author Response

Author's Notes to Reviewer 3:Thank you very much for the reviewer's affirmation. We double checked the manuscript, corrected some spelling mistakes, and revised or added some sentences according to the suggestions of other reviewers.

Reviewer 4 Report

The authors  found that the deletion of 18 minC, minD and minCDE resulted in enhanced hrpB1 and hrpF expression, which is dependent on 19 two key hrp regulators HrpG and HrpX. Yet, mutation of minC in Xoo resulted in significantly impaired virulence in host rice, and enhanced biofilm formation.The Xoo Min system is encoded by the minC, minD, and minE genes and the T3SS expression of Xoo  through HrpG-HrpX regulatory pathway and the involvement of the Min system in Xoo cell division, full virulence, swimming motility and biofilm formation followed by transcriptomic analysesThe authors described in detail their experimental protocol by construction of deleted mutants and complementation strains and Synteny Analysis on Chromosomes followed by microscopy analyses,Biofilm formation assay , Swimming motility assay,  Growth Measurement and Virulence assay of Xoo,and RNA-Sequencing and Real-Time Quantitative RT-PCR (qRT-PCR) Analysis. Western and Southern blotting analysis as well as GUS assays were performed followed by statistical analysis evaluation.

It is a scientifically sound paper, the authors showed that the Min system participates in the negative regulation of T3SS expression in Xoo, furtherly confirmed by the RNA-seq data,and provides information on the  Xoo virulence mechanism.

The paper is well organized and the bibliography is up to date.

My suggestion is to ACCEPT and publish it in its present form

Author Response

Author's Notes to Reviewer 4:Thank you very much for the reviewer's affirmation. We double checked the manuscript, corrected some spelling mistakes, and revised or added some sentences according to the suggestions of other reviewers.

Reviewer 5 Report

Revision of the Manuscript ID: microorganisms-1812097 entitled “The MinCDE Cell Division System Participates in the Regulation of Type III Secretion System (T3SS) Genes, Bacterial Virulence and Motility in Xanthomonas oryzae pv. oryza

The work presented here is of great interest in field of phytopathology, further elucidating the role of the Min system in the regulation of the Xoo T3SS, one of the fundamental virulence regulators in phytopathogenic bacteria.

The introduction is robust and well written, the methods detailed, clear and appropriated. Furthermore results are presented in a clear way. Discussion although good, I believe it should further explore the results obtained and try to end it with a correlation of all the findings in a final paragraph.

Some small typos were found, such as, final commas missing in the introduction, please double check entire document.

Author Response

Author's Notes to Reviewer 5: We gave a Conclusions in Part 4 that contained the elements (all important results ) the reviewer required. We double checked the manuscript, corrected some spelling mistakes, and revised or added some sentences according to the suggestions of other reviewers.

In figure1A, we corrected minC to minD.